

# Combined effects of water temperature, grazing snails and terrestrial herbivores on leaf decomposition in urban streams

Hongyong Xiang[1,2], Yixin Zhang[3], David Atkinson[2] and Raju Sekar[4]

[1] Department of Health and Environmental Sciences, Xi'an Jiaotong-Liverpool University, Suzhou, Jiangsu, China
[2] Institute of Integrative Biology, University of Liverpool, Liverpool, United Kingdom
[3] Research Center of Environmental Protection and Ecological Restoration Technology, Gold Mantis School of Architecture, Soochow University, Suzhou, Jiangsu, China
[4] Department of Biological Sciences, Xi'an Jiaotong-Liverpool University, Suzhou, Jiangsu, China

## ABSTRACT

The decomposition of organic matter in freshwaters, such as leaf litter, can affect global nutrient (e.g., carbon) cycling. This process can be influenced by fast urbanization through increased water temperature, reduced aquatic diversity and changed leaf litter quality traits. In this study, we performed a mesocosm experiment to explore the individual and combined effects of warming (8 °C higher and ambient), the presence versus absence of grazing snails (*Parafossarulus striatulus*), and intraspecific difference of leaf litter quality (intact versus > 40% area of *Liriodendron chinense* leaves grazed by terrestrial insects) on litter decomposition in urban streams. Litter decomposition rates ranged from $0.019 \, d^{-1}$ to $0.058 \, d^{-1}$ with an average decomposition rate of $0.032 \pm 0.002$ $d^{-1}$. All the three factors had significant effects on litter decomposition rate. Warming and the presence of snails accelerated litter decomposition rates by 60% and 35% respectively. Litter decomposition rates of leaves damaged by terrestrial insects were 5% slower than that of intact leaves, because litter quality of terrestrial insect-damaged leaves was lower (i.e., higher specific leaf weight) than intact leaves. For treatments with snails, warming stimulated microbial and snail mediated litter decomposition rates by 35% and 167%, respectively. All combinations of treatments showed additive effects on litter decomposition except for the interaction between warming and snails which showed positive synergistic effects. In addition, neither temperature nor litter quality affected snail growth rate. These results imply that higher water temperature and the presence of abundant snails in urban streams greatly enhanced litter decomposition. Moreover, the effect of pest outbreaks, which resulted in lower litter quality, can cascade to aquatic ecosystems by retarding microbe-mediated litter decomposition. When these factors co-occurred, warming could synergistically interact with snails to speed up the depletion of organic matter, while the effect of leaf quality on litter decomposition may be diminished at high water temperature. These effects could further influence stream food webs and nutrient cycling.

Corresponding author
Yixin Zhang,
yixin.zhang2019@suda.edu.cn

## INTRODUCTION

Global temperature is projected to increase 2.0–4.9 °C by the end of this century (*Raftery et al., 2017*), and the change in thermal conditions can influence almost all levels of stream ecosystems (*Daufresne, Lengfellner & Sommer, 2009*; *Woodward, Perkins & Brown, 2010*). Numerous studies have indicated that warmer water can accelerate leaf litter (hereafter litter) decomposition in streams (*Ferreira & Chauvet, 2011*; *Ferreira & Canhoto, 2015*; *Griffiths & Tiegs, 2016*; *Martins et al., 2017*). For example, from a synthesis of 1,025 records of litter decomposition, *Follstad Shah et al. (2017)* found that litter decomposition rates in freshwater ecosystems are expected to accelerate by 5–21% when water temperature increases 1–4 °C. By contrast, in a global spatial field experiment, *Boyero et al. (2011)* found that warmer conditions stimulated microbe-mediated litter decomposition whereas invertebrate-mediated litter decomposition was decreased. Consequently, overall litter decomposition rate was unchanged. However, these results may not be suitable for the projection of warming effects on litter decomposition in urban streams, because data from these two global-scale studies were collected from streams with low human-impact intensity. Differences in invertebrate and microbial communities, physical conditions, and other factors were associated with different responses of litter decomposition to water temperature between urban and non-urban (e.g., forest) streams (*Imberger, Walsh & Grace, 2008*; *Iñiguez Armijos et al., 2016*; *Wenisch et al., 2017*). For instance, dominant invertebrates in urban streams have broader thermal breadth than invertebrates in mountain forest streams (*Giersch et al., 2017*). Consequently, warming results in reduced abundance and richness of warming-sensitive invertebrates in forest streams–which are mainly responsible for macroinvertebrate-mediated litter decomposition (*Winterbourn et al., 2008*; *Griffiths & Tiegs, 2016*). By contrast, warming-induced reduction of abundance and richness of temperature-sensitive invertebrates in urban streams can benefit thermally tolerant invertebrates such as snails—the dominant contributor of macroinvertebrate-mediated litter decomposition (*Yule et al., 2015*). Therefore, results from most studies that have investigated warming effects on litter decomposition in non-urban streams may not be suitable for urban streams. Urban stream water temperature can be increased through various ways such as deforestation, water intake, discharging warmer effluent from domestic, industrial, and sewage-treatment sources (*Lepori, Pozzoni & Pera, 2015*), runoff from hot impervious surfaces and stormwater (*Somers et al., 2013*). Furthermore, as natural stressors usually interact with each other, the effects of warming on litter decomposition are also subjected to seasonal change (*Dossena et al., 2012*; *Ferreira & Canhoto, 2014*), nutrient concentration (*Ferreira & Chauvet, 2011*), and the presence of shredders (*Domingos et al., 2015*; *Moghadam & Zimmer, 2016*). Thus, predicting the consequences of warming on litter decomposition in urban streams needs to take account of other environmental stressors affecting consumer communities and litter quality.

In addition to abiotic factors (e.g., warming), biotic factors such as the presence of aquatic invertebrates (detritivores) can also affect litter decomposition (*Jonsson, Malmqvist & Hoffsten, 2001*; *Gonçalves, Graça & Callisto, 2006*). Shredders are usually recognized as the dominant contributor to invertebrate-associated litter decomposition in streams

(*Bruder et al., 2014*; *Taylor & Andrushchenko, 2014*). In many tropical and urban streams where shredders were scarce, litter decomposition rates did not differ between coarse and fine mesh bags (i.e., when invertebrates excluded (*Pascoal et al., 2005*). However, these studies may underestimate the role of scrapers in litter decomposition, which shear off food, especially periphyton adhered to leaf surfaces (*Cummins & Klug, 1979*). Specifically, many researchers have found a positive relationship between snail abundance and litter decomposition rate in streams where diversity and abundance of shredders are low (*Suren & McMurtrie, 2005*; *Chadwick et al., 2006*; *Yule et al., 2015*). Snails can colonize litter rapidly even before microbes (e.g., fungi) can develop sufficient biomass or partially degrade the leaf tissues (*Casas et al., 2011*). Snails can completely eat the soft part of the leaves (*Tanaka, Ribas & de Souza, 2006*). The presence of snails is likely to affect the microbe-mediated litter decomposition through: (1) changing competition in microbial communities via direct consumption of some microbes such as bacteria; (2) altering microenvironments on the leaf surface due to feeding activities; and (3) stimulating fungal growth by excreting nutrients (e.g., higher ammonia) and labile carbon (*Moghadam & Zimmer, 2016*) or decreasing turbidity which may influence the periphyton biomass (*Hann, Mundy & Goldsborough, 2001*; *Li, Liu & Gu, 2008*). Snails are abundant in urban streams due to their capability of tolerating high water temperatures and decreased water quality (*Gray, 2004*; *Ramírez et al., 2009*). In addition, dams can transform upstream reaches from lotic to lentic habitats in rural streams, thereby altering water depth, flow velocity, sediment and water temperature regime (*Stanley et al., 2002*; *Yan et al., 2011*; *Claeson & Coffin, 2016*). Although these changes may adversely affect sensitive invertebrates, other organisms including snails could benefit from these modified habitats (*Cross et al., 2010*; *Gangloff et al., 2011*). Therefore, snails may be an important factor influencing litter decomposition in these ecosystems and compensate for the loss of sensitive shredders (*Chadwick et al., 2006*; *Casas et al., 2011*).

Leaf quality has long been acknowledged as an important biotic factor influencing litter decomposition in streams (*Leroy & Marks, 2006*; *Hladyz et al., 2009*). Generally, high quality leaves (e.g., high nitrogen concentration) are more preferred by invertebrates and microbes, thus making them decompose faster than low quality leaves (*Schindler & Gessner, 2009*). Although numerous studies have investigated the effects of litter quality on its decomposition in freshwaters, most of them focused on interspecific differences in litter quality (*Leroy & Marks, 2006*; *Kominoski et al., 2007*; *Hladyz et al., 2009*) rather than intraspecific differences (*LeRoy et al., 2007*; *Jackrel, Morton & Wootton, 2016*). Environmental and anthropogenic stressors (e.g., insect herbivores, $CO_2$ concentrations, salt, and land use change) can decrease litter quality through various ways such as increasing concentrations of secondary chemicals (e.g., condensed tannins) (*Frost et al., 2012*; *Rothman et al., 2015*; *Jackrel & Morton, 2018*). For example, plants in warmer urban areas are likely to suffer more serious insect pest outbreaks than in cooler areas (*Meineke et al., 2013*), and insect herbivores may continue to increase in the future due to global climate warming (*Meineke et al., 2018*). According to the nutrient acceleration hypothesis, insect damage enhances litter decomposition due to accelerated senescence, increased nutrient cycling, and improved litter quality (*Chapman et al., 2003*). By contrast, as per the

nutrient deceleration hypothesis, insect attack can induce higher proportion of secondary defensive compounds which result in decreased nutrient cycling rates (*Schweitzer et al., 2005*), and thereby reduce litter decomposition rates. The changes in litter quality induced by insect herbivory can cascade to aquatic ecosystems (*Jackrel & Wootton, 2015*; *Jackrel & Morton, 2018*). Therefore, the effects of warming and other stressors associated with urbanization should be coupled with the effects of intraspecific litter quality to accurately estimate their individual and combined effects on ecosystem functioning and organism community structures (*LeRoy et al., 2007*; *Lecerf & Chauvet, 2008*; *Jackrel & Morton, 2018*). Such combinations of stressors are especially pertinent in urban and mountain areas (high elevation) where terrestrial insects are estimated to cause greater damage to plants, which could result in larger differences in intraspecific litter quality (*Chen et al., 2009*; *Meineke et al., 2013*; *Ramsfield et al., 2016*).

In this study, we used a mesocosm manipulative experiment to explore the effects of increased water temperature (∼8 °C) on the decomposition of intact and insect-damaged (>40% leaf area were grazed by insects) tulip poplar (*Liriodendron chinense*), in the presence and absence of snails (*Parafossarulus striatulus*). Specifically, we aimed to test whether: (1) increasing water temperature would accelerate both microbe- and snail-mediated litter decomposition; (2) leaf damage caused by terrestrial herbivorous insects would reduce litter quality and result in retarded litter decomposition rate; and (3) the presence of snails would increase litter decomposition rate and partly compensate for the loss of shredders.

## MATERIALS & METHODS

### Leaf litter collection

Freshly fallen *L. chinense* leaves were collected during the period 15th September to 1st October 2016 from a riparian forest in Jiulongfeng Nature Reserve (mean annual precipitation and air temperature are 1,500–1,600 mm and 15.4 °C, respectively), Anhui Province, China (30°6′39″N, 118°1′21″E). This *ca* 30-year-old *L. chinense* woodland had suffered from pests (mainly Lepidoptera: moths, Figs. S1A–S1B) since 2009. Terrestrial pest outbreak occur twice (June and September) every year. Consequently, the leaf-fall pattern had changed from once at the end of October to twice every year. In the laboratory, leaves were visually grouped into two categories (Fig. S1) according to the ratio of leaf area grazed by insects, i.e., intact-lightly damaged (0–5%) and heavily damaged (>40%). Then, leaves were oven dried (60 °C, 48 h) and weighed prior to use.

### Experimental design

Using ∼60 L aquaria (50 × 30 × 40 cm), factorial combinations of manipulated water temperature (ambient versus 8 °C above ambient), intraspecific leaf quality (intact versus insect-damaged) and scraper (snail) presence versus absence were each replicated five times (2 × 2 × 2 × 5 = 40 mesocosms). The experiment lasted for 25 days from 20th December 2016 to 14th January 2017. Mesocosms (Fig. S2) were installed on the riparian zone of a stream next to Xi'an-Jiaotong Liverpool University (31°16′30″N, 120°43′59″ E), Suzhou, China. Water was pumped from the stream and circulated within the mesocosms to emulate the natural water quality, flow, and microbial supply. There was no substrate (e.g., rocks,

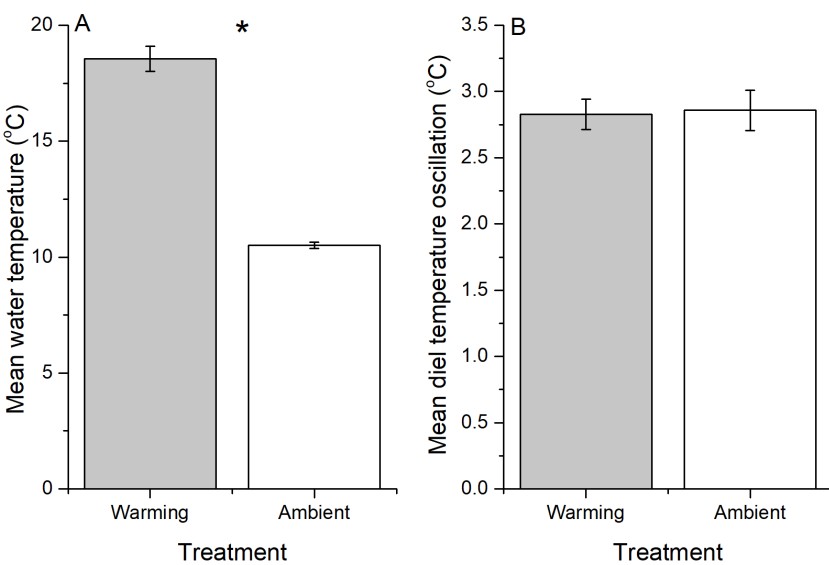

**Figure 1** **Averages of water temperature in ambient and warming treatments.** (A) Water temperature. (B) Diel temperature oscillation. Values are mean ± SE. The symbol * above the bar indicates a significant difference between the treatments.

gravel, sand) in the mesocosms. In each mesocosm, $10.01 \pm 0.01$ g of unconditioned litter (i.e., dry leaves) were added at the beginning of the experiment. One heating rod (LRB-210, 220–240 V, 100 W, SunSun Co. Ltd, Guangdong, China) was used to heat up the water in the warming treatment. The working temperature for the heating rod is 18–34 °C, as average annual air temperature in the study area (Suzhou, China) is 15–17 °C. We set the working temperature as 18 °C for each heater. In addition, warming usually induced a change of diel temperature oscillation, which can affect litter decomposition (*Dang et al., 2009*; *Vyšná et al., 2014*). Therefore, the diel temperature oscillation was also calculated. The average increased water temperature was 8 °C (Fig. 1 and Fig. S4), which is higher than the projected range of temperature increases by the end of this century globally (2.0–4.9°C) (*Raftery et al., 2017*) and for China (3.9–6.0 °C) (*Ding et al., 2006*). This extreme high temperature may still be possible in urban areas where air temperature could be 1−3 °C warmer than rural areas, and the difference in air temperature between urban and rural areas can be as large as 10 °C under certain conditions (e.g., calm, cloudless nights in winter) (*Grimmond, 2007*). In addition, for big cities such as Shanghai, China, the projected increase of mean temperature is estimated to be 2.5 times of that for global mean temperature (*Chu, Qiu & Xu, 2016*). As water temperature in most streams would increase 0.6–0.8 °C for every 1 °C increase in air temperature (*Morrill Jean, Bales Roger & Conklin Martha, 2005*), an 8 °C increase in water temperature would be realistic for streams in big cities. Moreover, an increase of 8 °C is not rare for laboratory microcosm studies (*Ferreira & Chauvet, 2011*; *Fernandes et al., 2012*; *Geraldes, Pascoal & Cássio, 2012*). Nine temperature loggers (ONSET, Stow Away TidbiT Temp Logger) were randomly placed into nine mesocosms (five warmed; four at ambient water temperature) to record
water temperature once every hour during the experimental period Dataset S1 and Fig. S3). Dissolved oxygen (DO), conductivity, and ammonia were measured using a YSI (Pro Plus), and pH and turbidity were measured using pH (CLEAN, PH30) and turbidity (HACH, 2100Q) meter, respectively, before conducting the experiment, and then on days 5, 14, and 24. On day 25, the litter (Fig. S5) was collected using a hand-held net, oven dried (60 °C, 48 h), and then weighed to calculate litter decomposition rate.

## Water temperature and water quality

Water temperature was successfully increased in mesocosms with the warming treatment (mean ± SE, 18.5 ± 0.54 °C) by an average of 8 °C above that in ambient treatment mesocosms (mean ± SE, 10.5 ± 0.14 °C) during the experimental period ($t = 14.537$, $df = 4.502$, $P < 0.001$, Fig. 1A and Fig. S1 in supporting information). However, diel temperature oscillation did not differ between warming (2.83 ± 0.24 °C) and ambient (2.86 ± 0.10 °C) treatments ($t = -0.124$, $df = 5.274$, $P = 0.906$, Fig. 1B).

Warming significantly affected all measured water quality variables (Table 1 and Table S1 and Data S2 in supporting information, Fig. 2), and such pattern was also found in other studies (*Domingos et al., 2015*; *Ferreira, Chauvet & Canhoto, 2015*;*Martínez et al., 2014*). Both pH and conductivity increased with increasing water temperature, while turbidity, dissolved oxygen (DO), and ammonia were reduced in warming treatments. The presence of snails decreased pH, turbidity, and DO. By contrast, snails increased ammonia and had no significant effect on conductivity. Litter quality only significantly affected pH by increasing pH in mesocosms containing insect-damaged litter. Most two-way interactions showed additive effects on water quality variables and only turbidity was affected by the three-way interaction.

## Specific leaf weight

Specific leaf weight (SLW, leaf dry weight per unit leaf area) can be regarded as an indicator of leaf toughness—an important litter quality trait (*Steinbauer, 2001*). SLW was measured to test the potential physical structural quality differences between insect damaged and intact litter, because plants tend to have higher SLW when attacked by insects (*Nabeshima, Murakami & Hiura, 2001*; *Sudderth & Bazzaz, 2008*). Thirty intact and insect-damaged leaves were randomly selected from the leaves collected for this study. For each focal leaf, one leaf disc (six mm in diameter, avoiding leaf vein) was randomly cut out using a core borer. All leaf discs were oven dried (60 °C) to constant weight, which was recorded to the nearest 0.0001 g (Dataset S3). Then, SLW was calculated by dividing the dry leaf mass to leaf disc area (*Jackrel, Morton & Wootton, 2016*).

## Snail

Specimens of a common snail *P. striatulus* were collected from a stream to the north of Renmin University of China, Suzhou (31°16′54″N, 120°44′30″E). This stream is straight, ~15 m wide, with muddy sediment and concrete bank. Snails were kept in mesocosms for at least one week to acclimate to the new environment and were starved for three days before conducting the experiment. Before starting the experiment, each snail was blotted dry and weighed to the nearest 0.0001 g (mean ± S.E, 1.0744 ± 0.0322 g, $n = 400$, (Dataset S4).
**Table 1** **Summary results of three-way ANOVA with repeated measures for the effects of water temperature (T), snail (S), and litter quality (Q) on water quality in experimental mesocosms.** Significant main effects are classified directionally as positive (+) or negative (−). Combined (C) two-way interaction effects are classified directionally (+ or −) as antagonistic (A), synergistic (S), additive (AD; no interaction) or no significant effect of either stressor (O) according to the conceptual model proposed by *Piggott, Townsend & Matthaei (2015)*. *P*-values <0.05 are in bold . Effect sizes (partial eta squared values; range 0–1) are shown in parentheses for all cases where *P* <0.1.

| Dependent variables | Litter quality | Q | Temperature | T | Snails | S | Q × T | C | Q × S | C | T × S | C | Q × T × S |
|---|---|---|---|---|---|---|---|---|---|---|---|---|---|
| pH | **0.003** (0.248) | + | **<0.001** (0.686) | + | **<0.001** (0.344) | – | 0.149 | AD | **<0.001** (0.402) | −S | 0.2 | AD | 0.215 |
| Turbidity (NTU) | 0.313 | | **<0.001** (0.93) | – | **<0.001** (0.615) | – | 0.394 | AD | 0.383 | AD | **0.001** (0.318) | −A | **0.003** (0.244) |
| Conductivity (μs/cm) | 0.422 | | **<0.001** (0.904) | + | 0.356 | | 0.315 | AD | 0.475 | O | 0.476 | AD | 0.996 |
| DO (%) | 0.163 | | **<0.001** (0.891) | – | **0.027** (0.143) | – | 0.928 | AD | 0.227 | AD | 0.403 | AD | 0.092 (0.086) |
| DO (mg/L) | 0.684 | | **<0.001** (0.941) | – | 0.445 | | 0.454 | AD | 0.383 | O | 0.827 | AD | 0.228 |
| Ammonia (mg/L) | 0.325 | | **<0.001** (0.887) | – | **<0.001** (0.492) | + | 0.252 | AD | **0.03** (0.14) | +S | **0.007** (0.206) | +A | 0.7 |

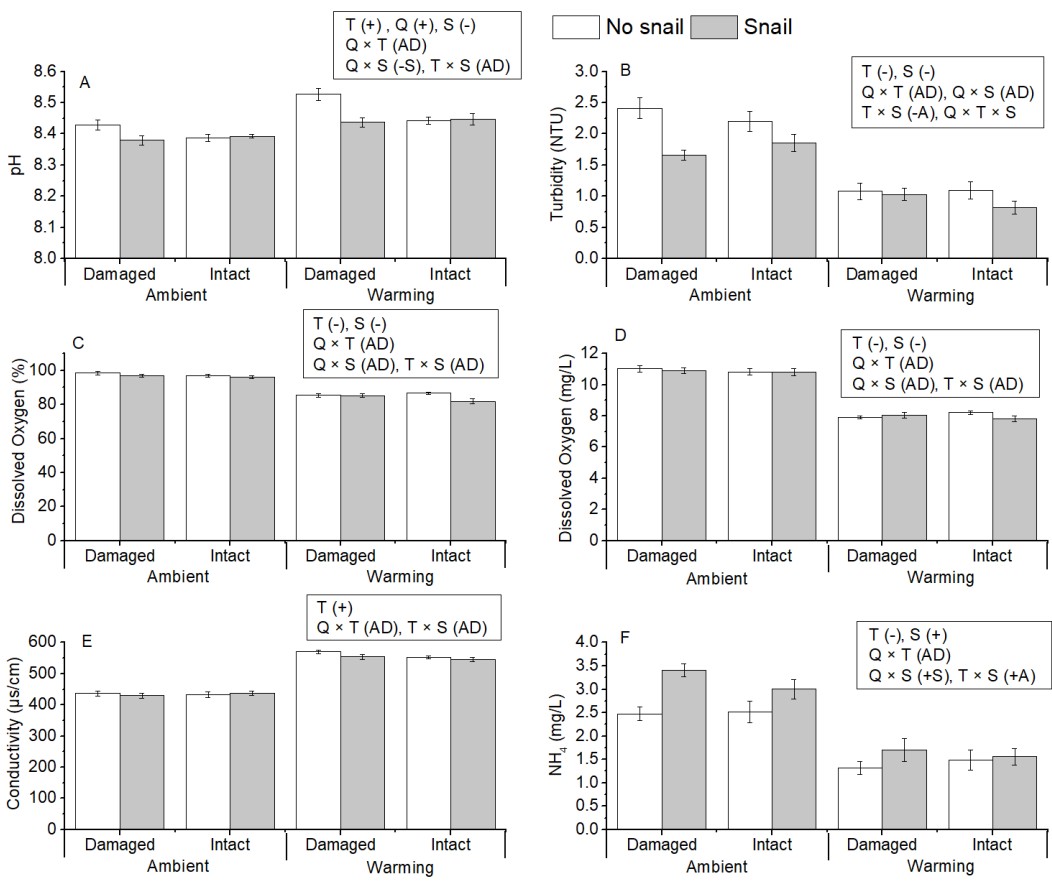

**Figure 2** Averages of measured water quality: (A) pH, (B) turbidity, (C) dissolved oxygen (%), (D) dissolved oxygen (mg/L), (E) conductivity, and (F) ammonia across the experimental treatments (water temperature: ambient and warming, snails: presence/absence, litter quality: intact and insect damaged). Values are mean ± SE (data of three sampling dates are combined). Text in rectangles indicates significant directional main effects and two-way interaction effects (water temperature: T, Snails: S, litter quality: Q), with effect classifications (for abbreviations see Table 1) in parentheses.

Twenty randomly selected snails were placed in each scraper treatment, giving a density of 133 ind/m$^2$, which was higher than the mean natural density (41–80 ind./m.$^2$) but still within the natural range of population density (0–280 ind./m$^2$) in this area (*Wang & Hong, 2010*; *Hu et al., 2013*). At the end of the experiment, snails were blotted dry and weighed again to determine the growth rate. Then, all snails were released to the stream where they were collected. Snail growth rates were calculated as $\mu = [\ln(W_t) - \ln(W_0)]/t$, where $W_t$ and $W_0$ were blotted mean wet mass per treatment at the end of the experiment (day t) and before the experiment, respectively (*Hill, Smith & Stewart, 2010*). In addition, snail tissue dry mass (TDM) and ash free dry mass (AFDM) were calculated using empirical equations in the study area (i.e., Suzhou, China), TDM = 0.067W, and AFDM = 0.286W, where W was blotted dry mass (*Zhao et al., 2009*).

## Litter decomposition

Litter decomposition rates (Data S5) were calculated assuming an exponential decay

$$(k, d^{-1}), W_{\text{t}} = W_I \times e^{-kt} \tag{1}$$

where $W_t$ represents the remaining leaf mass at the incubation time t (day) and $W_I$ is the initial mass of leaf material (*Ferreira & Chauvet, 2011*). In addition, we calculated the sensitivities of litter decomposition rates to temperature:

$$Q_{10-q} = (t_A/t_W)^{(10/(Tw-Ta))} \tag{2}$$

(*Conant et al., 2008*), where $t_A$ and $t_W$ are the time (days) to decompose 50% of initial dry leaf mass at the ambient and warming temperature respectively, $Ta$ and $Tw$ are the mean temperature during the experimental period in the ambient and warming mesocosms respectively. Litter decomposition rates in the presence and absence of snails was total ($k_{\text{total}}$) and microbe-mediated ($k_{\text{microbial}}$) litter decomposition rates respectively. The contribution of snail-mediated litter decomposition rate was estimated by the difference in dry leaf mass remains between mesocosms in the presence and absence of snails, and then calculating a new $k$ value ($k_{\text{snail}}$) (*Mosele Tonin, Ubiratan Hepp & Gonçalves, 2018*; *Magali, Sylvain & Eric, 2016*).

## Data analysis

Three-way ANOVA was used to test for differences in litter decomposition rates among treatments (warming, snail grazing, litter quality). In treatments with the presence of snails, two-way ANOVA was used to test the effects of warming and litter quality on litter decomposition rates, i.e., $k_{\text{total}}$, $k_{\text{microbial}}$, and $k_{\text{snail}}$.

$T$-tests were used to test for differences in water temperature (daily mean water temperature and diel temperature oscillation) between warming and ambient treatments (*Domingos et al., 2015*; *Ferreira, Chauvet & Canhoto, 2015*). For each measured water quality variable, we used three-way ANOVA (analysis of variance) with repeated measures (RM ANOVA) to explore the effects of experimental treatments on water quality. A $t$-test was used to check for SLW differences between intact and insect-damaged leaves. One-way ANOVA was used to detect whether initial blotted dry mass differed among the four treatments with snails. If they differed among treatments, then, initial blotted dry mass was set as a co-variable when doing the two-way ANCOVA to test the potential effect of body size on snail growth rate and litter decomposition rate. As both TDM and AFDM are correlated with blotted dry mass, we only analyzed two-way ANOVA results of net blotted dry mass growth rates. To determine the interaction type of two-way interactions, we followed the methods proposed by (*Piggott, Townsend & Matthaei, 2015*). After conducting normality tests for all data, the data were transformed (e.g., log) to improve the normality of data. All data were analyzed using SPSS 22.0.

## RESULTS

### Leaf litter decomposition

Litter decomposition rates varied between 0.019 d$^{-1}$ and 0.058 d$^{-1}$ (mean $\pm$ S.E, 0.032 $\pm$ 0.002 d$^{-1}$, Fig. 3, Fig. S3). Warming accelerated litter decomposition rates by

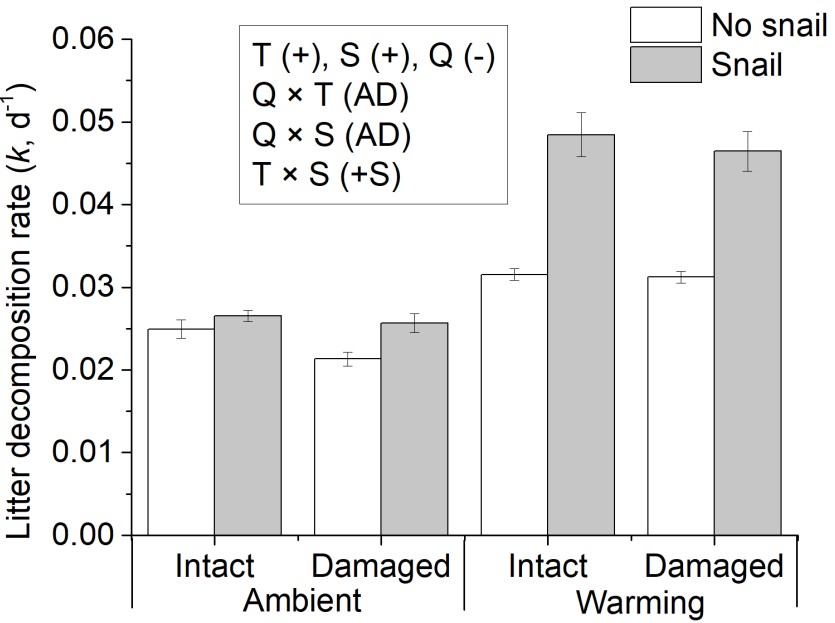

**Figure 3 Averages of litter decomposition rates.** litter decomposition rate ($k$, d$^{-1}$) for intact ($<5\%$) and damaged leaves ($>40\%$ leaf area were grazed by terrestrial insects) incubated in the absence (blank bar) and presence of snails (light grey bar), at ambient and warming mesocosms for 25 days. Text in rectangles indicate significant directional main effects and two-way interaction effects (water temperature: T, Snails: S, litter quality: Q), with effect classifications (for abbreviations see Table 1) in parentheses. Values are mean ± SE.

60% (data were log-transformed, $F_{1,32} = 259.93$, $P < 0.001$, Table 2). The presence of snails ($0.037 \pm 0.001$ d$^{-1}$) also significantly increased litter decomposition rates by 35% ($F_{1,32} = 90.21$, $P < 0.001$, Table 2). However, litter decomposition rates of terrestrial insect-damaged leaves ($0.031 \pm 0.001$ d$^{-1}$) were 5% slower than those of intact leaves ($F_{1,32} = 4.687$, $P = 0.038$, Table 2). The interaction of temperature and snail presence had positive synergistic effects on litter decomposition, i.e., warming increased the litter decomposition rates more in the presence of snails than in their absence. However, neither the rest of the two-way interactions nor the three-way interactions had significant effects on litter decomposition rates, i.e., the rest of the interactions all showed additive effects.

The overall sensitivity of litter decomposition rates to temperature ($Q_{10-q}$) was low according to a classification system reported previously (*Conant et al., 2008*). When mean temperature increased from 10.5 °C to 18.5 °C, the litter decomposition was stimulated more in the presence of snails than in their absence for both intact leaves ($Q_{10-q} = 2.38$ vs. 1.66) and insect damaged leaves ($Q_{10-q} = 2.37$ vs. 1.61). However, the thermal sensitivity of litter decomposition rates showed no difference between the intact and insect-damaged leaves in the presence ($Q_{10-q} = 2.38$ vs. 2.37) and absence ($Q_{10-q} = 1.66$ vs. 1.61) of snails.

In treatments with snails, warming significantly increased total, microbial-, and snail-mediated litter decomposition rates by 81%, 35%, and 167%, respectively ($P < 0.001$, Table 3, Fig. 4). Microbe-mediated litter decomposition rates were also 7% lower for

**Table 2 Summary (*P*-values and effect sizes) of three-way ANOVA comparing litter decomposition rates (exponential model) among experimental treatments (Litter quality: Q, water temperature: T, snails: S).** Main effects (M) are classified as positive (+) or negative (−). Combined two-way interaction effects (C) are classified directionally (+ or −) as antagonistic (A), synergistic (S), additive (AD; no interaction) or no significant effect of either stressor (O) according to the conceptual model proposed by *Piggott, Townsend & Matthaei (2015)*. *P* <0.05 are in bold print. Effect sizes (partial eta squared values; range 0–1) are shown in parentheses for all cases where *P* < 0.1.

| Treatments | df | Decomposition rate ($k$, d$^{-1}$) | | |
| --- | --- | --- | --- | --- |
| | | F | P | M/C |
| Q | 1 | 4.687 | **0.038** (0.128) | − |
| T | 1 | 259.930 | **<0.001** (0.890) | + |
| S | 1 | 90.210 | **<0.001** (0.738) | + |
| Q × T | 1 | 1.595 | 0.216 | AD |
| Q × S | 1 | 0.588 | 0.449 | AD |
| T × S | 1 | 25.503 | **<0.001** (0.444) | +S |
| Q × T × S | 1 | 1.808 | 0.188 | |
| Error | 32 | | | |

Notes.
Data of litter decomposition rates were log transformed to improve normality before conducting the analysis.

**Table 3 Summary (*P*-values and effect sizes) of two-way ANCOVA comparing the individual and combined effects of snail initial blotted dry mass (S), water temperature (T) and litter quality (Q) on total ($k_{total}$), microbe ($k_{microbe}$), and snail ($k_{snail}$) mediated litter decomposition rates.** Main effects (M) are classified as positive (+) or negative (−). Combined two-way interaction effects (C) are classified directionally (+ or −) as antagonistic (A), synergistic (S), additive (AD; no interaction) or no significant effect of either stressor (O) according to the conceptual model proposed by *Piggott, Townsend & Matthaei (2015)*. *P* <0.05 are in bold print. Effect sizes (partial eta squared values; range 0–1) are shown in parentheses for all cases where *P* <0.1.

| Treatments | df | $k_{total}$ | | | | $k_{microbe}$ | | | | $k_{snail}$ | | |
| --- | --- | --- | --- | --- | --- | --- | --- | --- | --- | --- | --- | --- |
| | | F | P | M/C | | F | P | M/C | | F | P | M/C |
| T | 1 | 113.558 | **<0.001** (0.883) | + | | 85.408 | **<0.001** (0.851) | + | | 30.008 | **<0.001** (0.667) | + |
| Q | 1 | 0.714 | 0.411 | | | 5.417 | **0.034** (0.265) | − | | 1.170 | 0.296 | |
| S | 1 | 0.179 | 0.678 | | | 0.650 | 0.433 | | | 0.016 | 0.902 | |
| T × Q | 1 | 0.001 | 0.982 | AD | | 4.194 | 0.058 (0.219) | AD | | 2.499 | 0.135 | AD |
| Error | 16 | | | | | | | | | | | |

damaged leaves than intact leaves ($F_{1,15} = 5.417$, $P = 0.034$, Table 3). By contrast, neither total nor snail-mediated litter decomposition rates were affected by litter quality. Water temperature and litter quality showed additive effects on each of the three measures of litter decomposition rate (Table 3). In addition, in the presence of snails, even though initial blotted dry mass differed among the four treatments ($F_{3,16} = 3.893$, $P = 0.029$, Fig. 5C), none of the litter decomposition rates were affected by initial dry mass of snails (Table 3).

## Specific leaf weight
Mean (±S.E.) SLW of terrestrial insect-grazed leaves (7.9 ± 0.4 mg/cm$^2$) were 47% higher than that of intact leaves (5.4 ± 0.4 mg/cm$^2$; $t = -4.872$, $df = 58$, $P < 0.001$, see (Dataset S3).

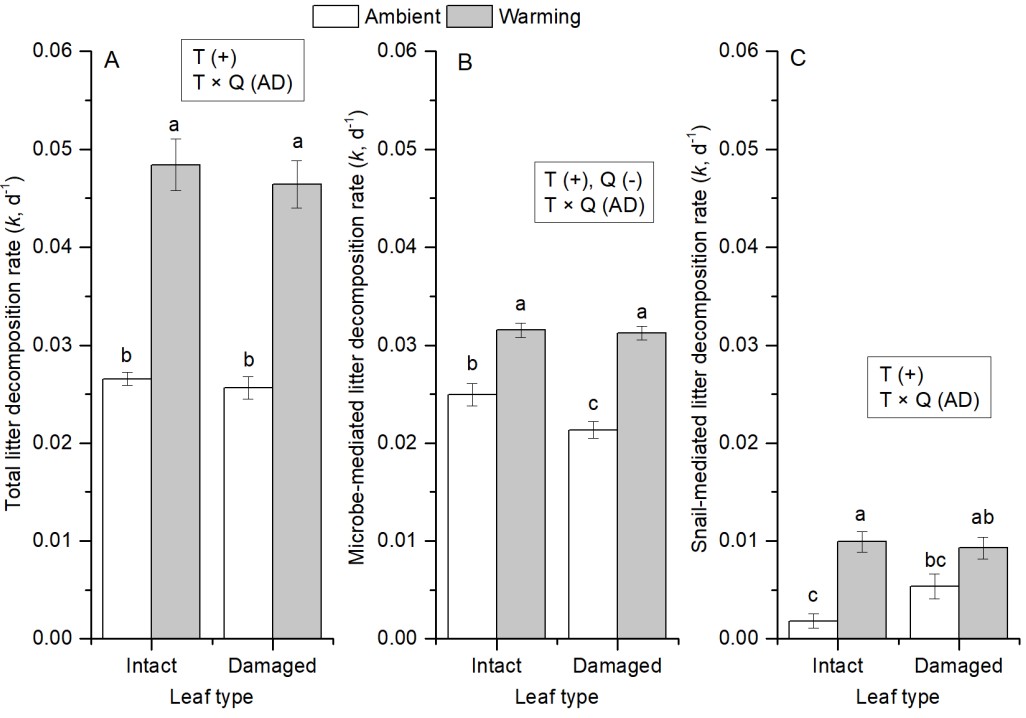

**Figure 4** Averages of litter decomposition rates of (A) total, (B) microbes, and (C) snails for intact and insect-damaged leaf litter at ambient and warming (∼8 °C higher) conditions. Different lowercase letters above each bar indicate significant differences after one-way ANOVA and *post hoc* Tukey (parameters with same letter are not significantly different between treatments). Text in rectangles indicate significant directional main effects and two-way interaction effects (water temperature: T, litter quality: Q), with effect classifications (for abbreviations see Table 1) in parentheses. Values are mean ± SE.

## Snail growth

There were no significant differences among treatments for snail growth rates (Table 4, Fig. 5A). Snail growth rates were negligible (close to 0, almost ceased growth) and net blotted dry biomass kept constant during the experiment (no significant difference was shown between initial and final blotted dry biomass).

## DISCUSSION

### Warming enhanced litter decomposition

Litter decomposition rates were significantly increased in warming treatments, which agrees with previous findings (*Martínez et al., 2014*; *Ferreira & Canhoto, 2015*), increasing by 7.5% per °C warming (mean $Q_{10-q} = 1.79$). This acceleration is nearer to the estimated 10% acceleration of litter decomposition rate per °C in the tropics rather than the 2.5% in temperate biomes (*Follstad Shah et al., 2017*). *Correa-Araneda et al. (2015)* indicated that warmer conditions can depress abundance and species richness of macroinvertebrates with narrow thermal tolerance thereby reducing litter decomposition rate. However, if the depressed macroinvertebrate-mediated litter decomposition was compensated by stimulated microbe-mediated litter decomposition, then overall litter decomposition

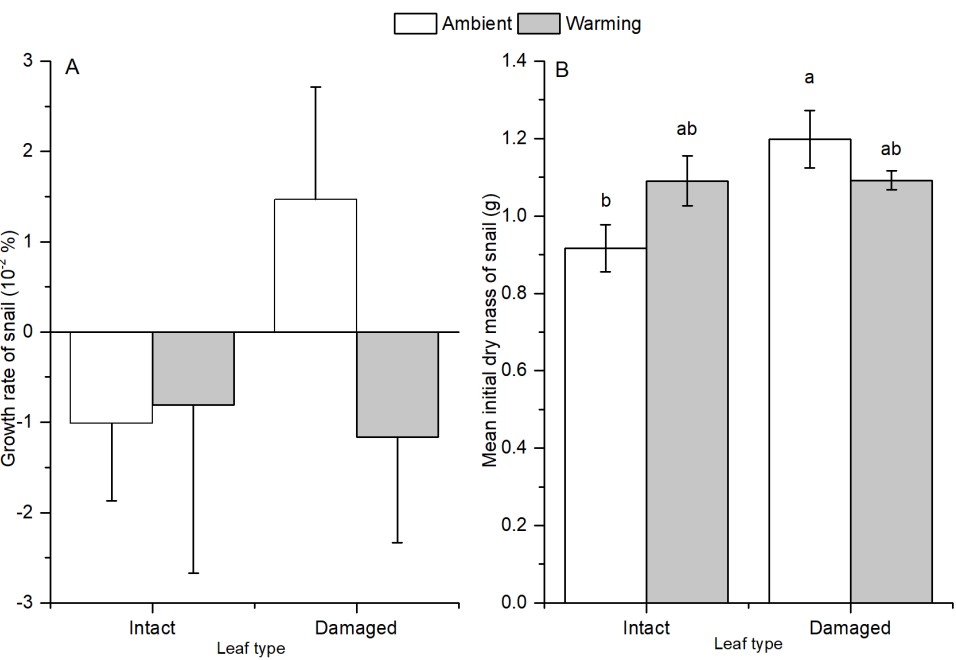

**Figure 5** **Averages of snail (A) growth rate, (B) initial blotted dry biomass in treatments of intact and insect-damaged litter at ambient and warming condition.** Values are mean ± SE. Different lowercase letters above each bar indicate significant differences after one-way ANOVA and *post hoc* Tukey (parameters with same letter are not significantly different between treatments).

would be unchanged (*Boyero et al., 2011*) or even higher than under ambient conditions (*Dossena et al., 2012*). Our results indicated that warming can not only enhance microbe-mediated litter decomposition (*Fernandes et al., 2012*; *Ferreira & Canhoto, 2015*), but also can accelerate snail-mediated litter decomposition (*Friberg et al., 2009*). Previous studies also found that microbe-mediated litter decomposition in urban streams was enhanced by the increased water temperature (*Imberger, Walsh & Grace, 2008*; *Yule et al., 2015*). However, in urban streams with poor water quality (e.g., low DO and high ammonia), microbe-mediated litter decomposition would decrease, countering any increase with warmer water temperature (*Martins et al., 2015*, *Iñiguez Armijos et al., 2016*). Therefore, it is important to take physicochemical factors into consideration when assessing the impact of increasing water temperature on litter decomposition in urban streams. For invertebrate-mediated litter decomposition, our result differs from that of *Domingos et al. (2015)* in which ∼3 °C higher water temperature depressed the activity of *Allogamus laureatus* (Trichoptera, Limnephilidae), leading to a lack of thermal stimulation of litter decomposition in the presence of *A. laureatus*. Thus, we suggest that differences in the invertebrate community can influence the effects of warming on invertebrate-mediated litter decomposition.

In addition, along with increased mean water temperature, higher diel temperature oscillation, which is usually associated with climate warming, can also contribute to accelerated litter decomposition rate (*Dang et al., 2009*; *Vyšná et al., 2014*; *Gonçalves, Graça*

**Table 4 Summary (*P*-values and effect sizes) of two-way ANOVA comparing snail growth rates for the individual and combined effects of litter quality (Q) and water temperature (T).** Main effects (M) are classified as positive (+) or negative (−). Combined two-way interaction effects (C) are classified directionally (+ or −) as antagonistic (A), synergistic (S), additive (AD; no interaction) or no significant effect of either stressor (O) according to the conceptual model proposed by *Piggott, Townsend & Matthaei (2015)*. *P* < 0.05 are in bold print. Effect sizes (partial eta squared values; range 0–1) are shown in parentheses for all cases where *P* < 0.1.

| Treatments | *df* | Snail growth rate | | |
|---|---|---|---|---|
| | | *F* | *P* | M/C |
| T | 1 | 0.881 | 0.363 | |
| Q | 1 | 0.187 | 0.671 | |
| Initial biomass | 1 | 0.201 | 0.66 | |
| T × Q | 1 | 0.434 | 0.52 | O |
| Error | 15 | | | |

*& Canhoto, 2015*). However, this was unlikely in this study, as diel temperature oscillation did not differ between warming and ambient treatments. Moreover, the effects of litter quality on both microbe- and snail-mediated litter decomposition rates diminished at higher water temperature, which accords with the suggestion of *Fernandes et al. (2012)* that warming (from 18 to 24 °C in their microcosms) could weaken the effects of litter quality on microbe-mediated (fungal) litter decomposition. Therefore, even though warming and other factors (e.g., increasing concentration of $CO_2$) associated with urbanization can alter litter quality (*Tuchman et al., 2002*; *Meineke et al., 2013*), these effects may be overridden by the effects of warming on litter decomposition in freshwaters.

## The presence of snails accelerated litter decomposition

The presence of scrapers (snails) accelerated litter decomposition rates by 35%. Our results suggest that the presence of snails in urban streams, where shredders are often scarce or absent, play an important role in litter decomposition, as also found by *Chadwick et al. (2006)*. We found that leaf morphology in treatments with snails differed from treatments without snails, indicating likely effects of grazing on the leaf surface. However, we could not ascertain whether leaf morphological changes arose from direct consumption of leaves by snails or from the indirect effects of grazing on algae attached to the leaf surface. The snails almost ceased growth in this study which was probably due to the overall low activities of snails in winter (*Eleutheriadis & Lazaridou-Dimitriadou, 2001*), or because we used unconditioned leaves and 25 days may not be long enough for sufficient colonization by microbes (*Wallace, Woodall & Sherberger, 1970*). Collectively, the presence of snails in urban streams can significantly accelerate litter decomposition in winter, even though snail growth rates were almost zero. Additionally, warming can synergistically enhance the effects of snails on litter decomposition. Due to the global increase of urban areas, we can anticipate that shredder-mediated litter decomposition would decrease while snail-mediated litter decomposition would be enhanced. In addition, water temperature would also likely increase as rural streams are transformed into urban streams, resulting in exaggerated snail-mediated litter decomposition.

## Terrestrial insect herbivores retarded litter decomposition

Decomposition rates of terrestrial insect-grazed leaves were 5% slower than those of intact leaves. The retarded litter decomposition was caused by lower litter quality, as indicated by (1) higher SLW of terrestrial insect-grazed litter (>40% leaf area) than intact leaves (<5% leaf area grazed), because a higher SLW has been associated with lower leaf N, P and N:P ratio (*Wu et al., 2012*), and (2) even though snail-mediated litter decomposition rates did not differ between intact and insect-damaged leaves, higher litter decomposition rates were found for insect-damaged leaves than for intact leaves under ambient conditions, because snails need to consume more insect-damaged leaves to compensate for the lower quality to meet their metabolic requirements (*Flores, Larrañaga & Elosegi, 2014*). These results suggest that insect herbivores decreased litter quality (*Peschiutta et al., 2018*), thereby supporting the nutrient deceleration hypothesis. Another possible mechanism is that insect herbivory resulted in higher concentrations of secondary compounds in deciduous trees (*Chapman, Whitham & Powell, 2006*). The lower litter quality had different effects on snails and microbes, with significantly slower microbe-mediated decomposition but faster (but not significant) snail-mediated decomposition in ambient conditions. This result is similar to that of *LeRoy et al. (2007)* in which aquatic fungi could discriminate intraspecific litter quality differences, whereas macroinvertebrates could not. *LeRoy et al. (2007)* suggested that aquatic fungi may respond to quality differences in litter. Snail-mediated litter decomposition showed a weak relationship to intraspecific litter quality difference, possibly because: (1) the effects of litter quality diffused through trophic levels, and (2) large body size of snails enable them to tolerate many toxicant secondary compounds (*Yule et al., 2009*). In addition, the differences in litter decomposition rates between insect-damaged and intact *L. chinensis* leaves in this study (5%) was much smaller than that of *Jackrel & Wootton (2015)* who observed 42% faster decomposition of intact *Alnus rubra* leaves than of herbivory-treated leaves (*Jackrel & Wootton, 2015*). The reason for this difference may be that litter quality of *L. chinensis* may be poorer than *Alnus rubra*: C:N stoichiometryof *L. tulipifera* (C:N, 36.69–56.3) (*Kominoski et al., 2007*; *Ardón, Pringle & Eggert, 2009*; *Griffiths & Tiegs, 2016*), may reflect that of the congeneric *L. chinensis*, and is higher (i.e., suggesting poorer quality) than that of *A. rubra* (C:N were 21.11 and 18.73 for herbivory treated and control respectively). Therefore, a further herbivore-induced decline in the already less palatable *Liriodendron* might not make a big difference for consumers. Although we only found increased pH in treatments with damaged leaves, other water quality characters may also have been potentially influenced by the difference of intraspecific litter quality (*Adams, Tuchman & Moore, 2003*), and consequently affect litter decomposition. Our findings imply that when considering the importance of litter quality on decomposition in streams, we should consider not only interspecific differences but also intraspecific differences in litter, especially considering that future climate change, land use change, and other stressors can change intraspecific litter quality (*Graça & Poquet, 2014*; *Fey et al., 2015*; *Pincebourde et al., 2017*).

## The interactions of water temperature, snail, and litter quality on litter decomposition

Among all the two-way combinations, only that between snails and water temperature showed positive synergistic effects on litter decomposition rates, whereas all other combinations showed additive effects (i.e., no significant interaction). The macroinvertebrate-warming synergistic effects on litter decomposition rates, also observed by *Moghadam & Zimmer (2016)*, could be explained by enhanced consumption rates of litter by snails at higher temperature conditions because of higher metabolic demands of snails at these high temperatures (*Seuffert, Burela & Martín, 2010*; *Gordon et al., 2018*). Warming can increase the community-level energy demand with consequences for ecosystem functioning (*Nelson et al., 2017*). At higher water temperatures, snails often feed more selectively on higher quality food (*Gordon et al., 2018*). This could be the reason why snail-mediated litter decomposition was more sensitive to temperature for intact leaves than for insect-damaged leaves. In addition, litter quality would be increased (e.g., reduced toughness, fewer phenols and lower C:N ratios) when incubated at higher water temperature (*Mas-Martì et al., 2015*). Interactive effects of litter quality and water temperature on litter decomposition are difficult to predict because of conflicting types of interaction including warming either reinforcing poor litter-quality effects on decomposition (*Correa-Araneda et al., 2015*; *Mas-Martì et al., 2015*), dampening the effects of lignin-rich (i.e., poor quality) litter on decomposition (*Fernandes et al., 2012*), or additively interacting with litter quality to affect litter decomposition (*Correa-Araneda et al., 2017*). Our results supported a disappearance of the effects of litter quality on litter decomposition at high water temperatures. These results imply that even though the presence of snails can increase litter decomposition in urban streams, most of the carbon stored in litter is released by microbes and transformed into $CO_2$. Decomposition of lower quality litter is expected to be more stimulated by microbes than is higher quality litter at high water temperature condition. By contrast, snail-mediated litter decomposition may be more sensitive to the change of water temperature for high rather than low quality litter. Therefore, the effect of warming on nutrient cycling in urban streams depends on litter quality.

## Implications for urban stream management and conservation

Our results indicate that reducing the impacts of warming should be the most important way to alter organic matter decomposition in urban streams, rather than the other two factors (intraspecific litter quality difference and the presence of snails). Warming can also induce a change of DO concentration, terrestrial subsidy input (quality, quantity, and input time of litter), and macroinvertebrate and microbial communities, which are among the 26 key research questions in urban stream ecology (*Wenger et al., 2009*), and consequently affect nutrient cycling (e.g., carbon) in these waterbodies. The effects of warming on nutrient (e.g., carbon) cycling through litter decomposition in streams depend on how much of this carbon goes into invertebrates (invertebrates converted litter to particulate and dissolved forms of carbon) or microbes (microbes released the carbon stored in litter to gaseous form) (*Follstad Shah et al., 2017*; *Boyero et al., 2011*). This implies that $CO_2$ production via litter decomposition in urban streams might increase with warming, as

well as the generation of particulate or dissolved carbon. However, this projection would be improved if we were able to know how future climate change would affect the quantity and quality of litter, and macroinvertebrate communities in freshwaters. Considering that urban streams are expected to suffer more serious stress from warming than rural streams, it is urgent to take actions to alleviate their negative effects on freshwaters. In particular, given the predicted increase in urban land cover of 1.2 million $km^2$ by 2030, , which is three times the global urban land area in 2000 (*Seto, Güneralp & Hutyra, 2012*), more streams will clearly be affected by this land-use change. Conservation actions to mitigate the effects of climate warming on urban ecosystems include: (1) increasing urban forest cover by sequestering $CO_2$ (*Bowler et al., 2010*; *Barò et al., 2016*) and reducing storm water runoff (*McPherson et al., 1997*); (2) enhancing hyporheic exchange and adopting different wastewater treatment strategies through accelerated heat exchange with other media such as atmosphere and subsurface groundwater (*Kaushal et al., 2010*); (3) decreasing the quantity of water withdrawals by reducing the warming effects induced by impoundments (*Webb & Nobilis, 1995*) and (4) increasing the reuse of treated wastewater (*Kinouchi, 2007*).

## CONCLUSIONS

In summary, we found that (1) litter decomposition rates were stimulated by increasing water temperature (∼8 °C higher than ambient) through increased activities of microbes and invertebrates (snails); (2) the presence of grazing snails (scrapers) accelerated litter decomposition rate through their direct consumption of leaf material or indirectly by scraping microbes attached to leaf surfaces, and these effects were stronger at raised water temperature than at ambient water temperature; and (3) terrestrial herbivorous insects retarded microbe-mediated litter decomposition by inducing higher SLW of litter (i.e., poorer litter quality), and the effects of litter quality on both microbial and snail mediated litter decomposition diminished at higher water temperature. Thus, although the increasing terrestrial insect herbivory could lead to lower litter quality that can retard litter decomposition (*Adams, Tuchman & Moore, 2003*; *Meineke et al., 2018*), warming is expected to stimulate both microbe- and snail-mediated litter decomposition in urban streams.

## ACKNOWLEDGEMENTS

We thank Xiaoyan Ni, Ting Zhang and Ying Hua for their help in collecting leaf litter and snails and setting up the mesocosm experiment. We also thank for the valuable comments given by Dr. Alan Tonin and other anonymous reviewers which were very helpful in improving the quality of this manuscript.

### Funding

This work was supported by the Research Development Fund Project of Xi'an Jiaotong-Liverpool University (RDF-15-01-50) and the Natural Science Foundation of Jiangsu

Province (BK20171238). The funders had no role in study design, data collection and analysis, decision to publish, or preparation of the manuscript.

### Grant Disclosures

The following grant information was disclosed by the authors:
Research Development Fund Project of Xi'an Jiaotong-Liverpool University: RDF-15-01-50.
Natural Science Foundation of Jiangsu Province: BK20171238.

### Competing Interests

The authors declare there are no competing interests.

### Author Contributions

- Hongyong Xiang conceived and designed the experiments, performed the experiments, analyzed the data, contributed reagents/materials/analysis tools, prepared figures and/or tables, authored or reviewed drafts of the paper, approved the final draft.
- Yixin Zhang analyzed the data, contributed reagents/materials/analysis tools, authored or reviewed drafts of the paper, approved the final draft.
- David Atkinson analyzed the data, contributed reagents/materials/analysis tools, authored or reviewed drafts of the paper, approved the final draft.
- Raju Sekar analyzed the data, contributed reagents/materials/analysis tools, authored or reviewed drafts of the paper, approved the final draft.

### Data Availability

The raw data is available in the Supplemental Files.

### Supplemental Information

Supplemental information for this article can be found online at http://dx.doi.org/10.7717/peerj.7580#supplemental-information.

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
