# Peer review of "Combined effects of water temperature, grazing snails and terrestrial herbivores on leaf decomposition in urban streams"

_PeerJ, doi:10.7717/peerj.7580_

## Round 0.1 · original submission · Major Revisions

Please find below the comments by the two reviewers who have evaluated your manuscript. I have read the manuscript myself, and I generally agree with all the reviewers' comments, which should be thoroughly addressed if you decide to resubmit your manuscript to PeerJ. The manuscript shows interesting results that are potentially worth being published. However, as pointed out by the reviewers, the manuscript also shows some important caveats that need to be addressed before the manuscript can be considered for publication. Please pay particular attention to improving issues regarding the experimental approach and data analysis.

Reviewer 1 ·

Basic reporting

This paper reports an experiment examining leaf litter decomposition rates in mesocosms mimicking urban streams, at different levels of water temperature (10 vs. 18 ºC), grazing snails (presence vs. absence) and leaf damage (intact vs. damaged by terrestrial herbivores). The rationale behind this study is that (1) the above factors differ between non-urban and urban streams (the latter being warmer, with greater presence of grazing snails, and receiving more damaged litter), but (2) most available information about factors affecting litter decomposition comes from forest streams. While this is a valid argument and an interesting topic, the study presents important flaws, which I detail in the other sections below.

The text in general needs to be improved, and ideally checked by a native speaker. There are many cases where sentences are not concluded, or punctuation marks are not used properly. Verb tenses are often used incorrectly (e.g., L109, 114) and there are quite a few typos (e.g. “increasement”, “may be continue to increase”). Some paragraphs are too large, and consecutive paragraphs are sometimes poorly connected.

Experimental design

It is unclear whether the simulated warming responds to differences between non-urban and urban streams in the study area or to climate change. Some parts of the text seem to suggest that both impacts are addressed, while this is clearly not the case. So, first of all, the authors should be clear about what kind of impact they are studying (i.e., effect of urbanisation vs. effects of climate change on litter decomposition) and make it clear in the text. In fact, the temperature treatments used may or may not be realistic, depending on which impact they target. For example, while a difference of 8 ºC could easily occur between forest and urban streams, this difference would be too high if related to climate warming, based on projections for the end of this century, even under the worst scenario for China (a max. increase of 6.0 ºC, cited on L184).

Having said that, if effects of urbanisation are the main focus, then other issues arise. Urban streams differ from non-urban streams not only in terms of the factors manipulated in this study (i.e., temperature, invertebrates and litter inputs), but also, and very importantly, in terms of water quality and habitat complexity. Thus a proper experimental design comparing non-urban and urban streams should take these factors into account. This is obviously not the case here, as only one type of water was used (from an urban stream?) and habitat complexity was kept constant (no substrate). Moreover, if non-urban and urban streams are compared, but only grazing snails are taken into account, then the role of shredders in non-urban streams is underestimated.

The issues raised above suggest that the present design is more appropriate to test effects of climate warming than effects of urbanisation. However, if this is the case, it needs to be made clear, consistently, throughout the whole ms. Furthermore, the experimental treatments used need to be better justified, and more basic field information needs to be provided in order to support predictions. In relation to this, the hypotheses need refinement. For example, “increased temperature” should be used instead of “climate warming”. Any mention to leaf quality should be excluded, as it was not examined. Hypothesis 4 is not properly formulated; if synergistic effects are expected for any combination of factors, then this should be explained. In general, it should be made clear why hypotheses are novel and how they are based on previous evidence (ideally, providing references).

Validity of the findings

I wonder how damage by terrestrial herbivores actually altered leaf quality. By measuring N and P concentrations in intact and damaged leaves the authors would have been able to actually show that herbivory decreased leaf quality. Instead, they relied on the measurement of “specific leaf weight”, which only indicates a relationship between leaf mass and area. While this leaf trait is often related to leaf toughness and a good indicator of leaf palatability, at least for shredders, I’m not sure how this applies to leaves that have been grazed by terrestrial insects. What I mean is, if leaves present holes due to insect grazing, then the relationship between leaf mass and area is clearly affected, but that does not necessarily mean that the remaining leaf material is of worse quality than that of an intact leaf. Pictures of intact vs. damaged leaves might be helpful, but only pictures of leaves at the end of the experiments are provided.

The data analyses section is quite confusing. In order to improve clarity I suggest to split it into different paragraphs and, importantly, clarify which analyses were used to test each of the hypotheses. For example, it is unclear why two different measures of litter breakdown rate (k and % litter mass loss) were used; this should be justified, otherwise choose one of them and discard the other. It is also unclear why temperature sensitivity was examined. Lastly, the explanation about what additive and synergistic effects are seems superfluous.

·

Basic reporting

The study performed by Xiang and colleagues is exciting and provide many relevant informations regarding the impact of urban streams on a key ecosystem process, litter breakdown. However, my main concern is with multiple topics of data analysis, which compromises the quality of the work so far. On the other hand, the authors can overcome these issues by performing more adequate analysis and text reorganization. Also, the presentation of results (including figures and tables) can be improved by the synthesizing results description and prioritization of the core of the study (i.e., litter breakdown). I have provided several suggestions to improve this important work in the sections below.

Experimental design

Material and Methods: the ‘leaf litter collection’ section should be first than ‘experimental design’.

Lines 168-169: I never seem before this kind of methodology to estimate growth rate, i.e., blotting dry alive individuals before and after the experiment. You provide some references to backup this procedure or at least justify why common procedures such as size-to-body mass curve estimations (i.e., measures of size to estimate body dry mass of individuals; Smock 1980, https://doi.org/10.1111/j.1365-2427.1980.tb01211.x) were not used.
Also, explain in details how growth rate was calculated.

Line 172: describe mesocosms here and not on line 136 and also provide more details about mesocosms installation and maintenance. Photos and schemes of mesocosms distribution within the study site would be very helpful and illustrative (even in grey scale or at least in the supplementary material).
Line 175-176: 10.00 ± 0.01 g instead.
Line 176: replace ‘unconditioned’ by ‘dry’.

Data analysis
The data analysis section should report only the analytical procedures related to the exploration of your data and/or hypothesis testing but not formulas or any other calculation to obtain the final response variables; otherwise, this section would be too messy. Thus, I suggest you move the breakdown rate estimation, growth rates, snail tissue dry mass (note that it was the first mention!), etc to their respective sections above.
Please, use different paragraphs for different subjects such as main tests (those testing the study hypotheses), secondary tests (testing any assumption, i.e., ‘specific leaf weight differences between intact and damage leaves’) and other estimations (i.e., synergies among predictors).
Regarding the analyses performed, a Student’s t-test is not adequate to time-series data such as water temperature because it assumes independence of measures, which is not your case – i.e., as you recorded one value of temperature at each hour for the same mesocosms. An adequate analysis, for instance, is a repeated measure ANOVA or a linear model using a temporal correlation component (Quinn & Keogh 2002, Zuur et al. 2009)

Quinn, G. P., & Keough, M. J. (2002). Experimental design and data analysis for biologists. Cambridge University Press.
Zuur, A., Ieno, E. N., Walker, N., Saveliev, A. A., & Smith, G. M. (2009). Mixed effects models and extensions in ecology with R. Springer Science & Business Media.

Regarding the analyses of water parameters, first, I did not understand why they were conducted because you have not stated any hypothesis for it (i.e., there were not previous expectation to justify such analyses). Also, repeating several times the same analysis for different parameters (in this case, water parameters) is not wrong, but increase the probability of Type I statistical error. Additionally, most of those water parameters should be very correlated (DO in % and in mg/L; conductivity and turbidity and so on) and then would be difficult (or even impossible) to determine if a parameter have changed as a function of a treatment or conduced by any other parameter.
Litter mass loss and breakdown rates are intrinsically correlated as both use the same set of data with slightly differences in their estimates. Thus, there is no need to use both in data analysis. I suggest you maintain litter breakdown (and only report the % of litter mass loss in the results, excluding it from figures and tables also) to be consistent with the estimation of k-total, k-microbial and k-snails, which are very interesting and relevant to unravel some of the effects being tested here.
Regarding snail growth, the authors should test if there was actually any (statistical) difference between initial and final mass of snails (that is, if there was growth). Looking for Figure 5a (especially for standard errors) it seems there is no growth (i.e., higher variability that surpass the zero line).
Finally, a much more serious concern (than normality) for any kind of parametric analysis is the assumption of homogeneity of variances among the levels of a treatment (see Zuur et al. 2009, Chapter 2, for a deeper discussion). Transforming data is not always a guaranty to the adequacy of this assumption. I suggest you present boxplots using litter breakdown rates (as your main response variable) in relation to each treatment, as a supplementary material – as visual inspection of this assumption is a much more recommended approach than any other hypothesis testing approach.

Validity of the findings

Results
Water parameters: should be described in the methods, as they are description of conditions experienced at each treatment (and not related to any hypothesis being tested directly, as I pointed above). Also, it is obvious that temperature would affect parameters such as pH, conductivity, dissolved oxygen and turbidity, as all are temperature dependent; please, make this clear.

Specific leaf weight: should also be described in the methods, because this comparison ensures you indeed have used leaves with different ‘qualities’ in your experimental set-up.

Leaf litter decomposition: Importantly, when a significant interaction (e.g., temperature and snail presence) is detected it means you cannot interpret each individual factor per se, because they are dependent of each one. In this case you should only report the interaction and discuss it; please, review it throughout the text.
Would be more interesting and informative if you report averages ± SE for each level of a treatment (instead of a value for the whole treatment as you did in line 273), when a difference was detected.
Indicate the % or proportion of difference when a lower or higher rate was reported. For instance, “Microbe-mediated litter breakdown rates were also XX% lower than damage leaves …” (lines 288-290).

Lines 268, 270, 272 and so on: Do not use decimal places when presenting %, as will be easy to read.

Discussion
Warming enhanced litter decomposition: the authors could provide a better contextualization of this effect in urban streams (mainly comparing the present results with others in similar conditions or regions) and try to put their findings into a real world context.
Please remove the paragraph between lines 332-346 as it do not add anything novel or contribute to your work.
The presence of snails accelerated litter decomposition: Lines 349-354: It is a repetition of information already present in the introduction; please start this sub-section on line 355.
Line 355: add “where shredders are often scarce or absent” after ‘urban streams’ between commas.
Line 363: replace ‘normal’ by ‘sufficient’
Lines 371-373: this idea should be further developed, as it is one of the most interesting at all. In what conditions and land uses should be expected the greater changes?

Terrestrial insect herbivores retarded litter decomposition: The authors should pay special attention to the fact that while specific leaf weight can be negatively correlated with nutrient concentrations, it is also is with secondary compounds. It means leaves are one of two options: better defended physically (higher toughness) or chemically (higher content of toxic compounds), but not both (Boyero et al. 2007, https://doi.org/10.1038/s41598-017-10640-3)

The interaction of water temperature, snail and litter quality on litter decomposition: I found this section too speculative and with many low-relevance arguments, especially because this combined influence of three factors is of most relevance of disturbed streams, such as the urban ones. The authors could explore much better the idea of higher metabolic demands of snails and its implications. Also, the snail mediated litter breakdown indicates that intact leaves are more prone to temperature effects than damage leaves; the microbial mediated litter breakdown evidence the temperature effect on microbial litter processing (independent of leaf type) and give some clues about why snail litter breakdown are intensified with temperature (i.e., higher microbial development and thus, higher nutritive quality of litter). This should be explored further.

Additional comments

Please double space all the text as will be easier to read.

The fourth hypothesis is quite uncommon; if the authors do not have an a priori expectation/hypothesis (a specific one) regarding interactions among factors then you should assume that this is exploratory rather than formulating a general and counter-productive hypothesis.

Line 16: I suggest you start the abstract with a broader context sentence such as the importance of litter decomposition and the prominent conversion of natural to urban streams worldwide; this will attract greater interest to your study.

Line 24: remove ‘winter’ as it is detailed in the methods and may give the wrong idea to the readers that your results may be only applicable to this season (but of course, further studies to test this assumption may be needed).

Lines 27, 28 and 30: round decimal places; use integers for simplicity.

---

## Round 0.2 · Minor Revisions

Thanks for submitting your revised manuscript to PeerJ. In general, the authors have correctly incorporated the suggested changes. However, both reviewers and myself still have some concerns that need to be addressed before the manuscript can be accepted for publication. Please, pay attention to the very helpful suggestions made my reviewer 1 in the annotated manuscript mainly to improve clarity, language and style. Pay also attention to the technical issues indicated by reviewer 2.

[]

Reviewer 1 ·

Basic reporting

The authors have addressed my previous concerns and have significantly improved the writing. I have attached a document with further editing and a couple of minor comments.

Experimental design

No comment

Validity of the findings

No comment

Additional comments

No comment

Annotated reviews are not available for download in order to protect the identity of reviewers who chose to remain anonymous.

·

Basic reporting

The authors have done a good job in addressing my previous comments, however, there still are some issues which were not properly considered or further points that deserve attention (not raised before but mostly that appear only in this new version of the MS).

Experimental design

No comment

Validity of the findings

No comment

Additional comments

Lines 331-338 & 339-345: Should be moved to a subsection entitle ‘Litter breakdown’ before the data analysis part.

Lines 338-339: note that ‘litter mass loss’ was removed from the MS as the authors previously mentioned. Please, remove all mentions to this estimation throughout the text.

Lines 348-350: You should explain how Dataset 1 (which contains water temperature data) was aggregated to allow the authors to perform a t-test. The authors probably averaged temperature data over all the experimental period for each mesocosm, resulting in a total of ten values. This is acceptable, however, for diel temperature oscillations the same procedure were not performed – inferred based on the degrees of freedom of t-test on line 273 of the MS. To fix this issue, I suggest you first calculate daily temperature averages for each mesocosm, then calculate a measure of variability (standard deviation, variance or coefficient of variation over the experimental period) and finally test this variability using a t-test.

Lines 348-493: this information should go in the supplementary material, maybe in a section ‘Supplementary methods’ as the results of these analyses were previously reported above and are in support of the experimental setup but not a result/finding of the study.

Lines 524-525: this sentence is not correct, because you are referring to ‘treatments with snails’ but at the same time mention ‘total’, ‘microbial’ and ‘snail-mediated breakdown’, which all were estimated based on the difference between treatments with and without snails. Please, review it. For example, when you use total/microbial/snail breakdown rates you have to forget about treatments with and without snails and focus the comparison on warming and/or leaf damage occurrence.

Line 617: why did the authors use SLW in μg/mm2 instead of mg or g/cm2, which is a much more intuitive unit?

Figure 5. I suggest you delete the panel B (change of blotted dry mass), as this information is essentially the same as growth rate. Please, also remove any mention of change of blotted dry mass from the text for simplification.

Table 2. Initial litter mass loss should be delete, as indicated previously. Also, there is no need to log-transformed litter breakdown rates (to attain normality) as they are already very small numbers (< 0.1) – which suggest any transformation would not be efficient to this end but could remove important/interesting patterns in data. Thus, I suggest the authors to re-run this analysis.

---

## Round 0.3 · accepted · Accept

The supplementary methods section does not describe methods but rather results. I suggest you remove the supplementary methods from the paper and include a short version of the text included there in the results section where Fig. 2 is cited.

In addition, please recheck the paper, so that all figures and tables are correctly cited in the text and so that the number of typos is minimized. Here are some typos I found:

L189: "Figure" should be "Fig.").
L193: leave space after "(2015)"
L195: Remove ")".
L310: leave space after "laureatus"